# The Effects of High- Versus Moderate-Intensity Exercise on Fatigue in Sarcoidosis

**DOI:** 10.3390/jcm8040460

**Published:** 2019-04-05

**Authors:** Anita Grongstad, Nina K. Vøllestad, Line M. Oldervoll, Martijn A. Spruit, Anne Edvardsen

**Affiliations:** 1LHL Hospital Gardermoen, 2067 NO Jessheim, Norway; Anne.Edvardsen@lhl.no; 2Department of Interdisciplinary Health Sciences, University of Oslo, 0318 NO Oslo, Norway; n.k.vollestad@medisin.uio.no; 3LHL, Clinics Trondheim, 7041 NO Trondheim, Norway; Line.Merethe.Oldervoll@lhl.no; 4Department of Public Health and Nursing, Faculty of Medicine and Health Sciences, Norwegian University of Technology and Science (NTNU), 7491 NO Trondheim, Norway; 5Department of Research & Education, CIRO, Center of Expertise for Chronic Organ Failure, 6085 NM Horn, The Netherlands; martijnspruit@ciro-horn.nl; 6Department of Respiratory Medicine, Maastricht University Medical Centre, NUTRIM School of Nutrition and Translational Research in Metabolism, 6229 HX Maastricht, The Netherlands; 7REVAL—Rehabilitation Research Center, BIOMED—Biomedical Research Institute, Faculty of Rehabilitation Sciences, Hasselt University, 3590 BE Diepenbeek, Belgium

**Keywords:** pulmonary sarcoidosis, endurance training, high-intensity interval training, feasibility

## Abstract

Background: Fatigue is a common symptom in patients with sarcoidosis. Despite lacking evidence on whether high-intensity interval training (HIIT) will aggravate fatigue, moderate-intensity exercise is often recommended. This study aimed to investigate whether a single session of HIIT would affect fatigue differently from a single session of moderate-intensity continuous training (MICT). Methods: Forty-one patients with pulmonary sarcoidosis were recruited to a cross-over study. All patients completed one treadmill session of HIIT (85% of peak heart rate (HRpeak)) and one of MICT (70% of HRpeak). Fatigue was assessed with the Visual Analogue Scale 0–100 mm, before (T0), after (T1), and 24 hours after (T2) each exercise session. Paired sample t-test was used to compare changes in fatigue from T0 to T1 and from T0 to T2 between HIIT and MICT. Results: No statistically significant difference in fatigue levels was found between HIIT and MICT, either at T1 (3.6 (13.5) and 1.4 (13.5)) or at T2 (8.2 (17.0) and 2.1 (17.1)). Conclusions: A single session of HIIT did not affect fatigue differently than a single session of MICT. These preliminary findings support the need for further research on the long-term effect of HIIT on fatigue in patients with sarcoidosis.

## 1. Introduction

Sarcoidosis is a multisystem granulomatous disorder affecting any organ. The lung is involved in more than 90% of the patients [1]. Up to 80% of the patients with sarcoidosis report moderate to severe fatigue as one of the most disabling symptoms [2]. Sarcoidosis-related fatigue is a complex symptom reported by patients, and objective measurements such as lung function tests and chest radiography correlate poorly with patients’ perceptions of fatigue [3]. Patients with sarcoidosis have a reduced exercise capacity compared to healthy individuals [4]. Because of the complexity of fatigue, both patients and healthcare professionals express reservations in relation to exercise intensities and, in turn, the potential aggravation of fatigue. To our knowledge, only four studies have explored the effects of an exercise program on sarcoidosis-related fatigue in patients with pulmonary sarcoidosis [5,6,7,8]. They showed promising improvements in exercise capacity after three months, but the effects on fatigue are still inconclusive [9]. One of the studies reporting less improvement than expected in exercise capacity and fatigue thought this was due to the low intensity of the exercise programs considered [7], while the only study using high-intensity exercise showed a statistically significant improvement in fatigue and believed it was related to the increased exercise capacity [8]. There is good evidence that high-intensity interval training (HIIT) is more effective to improve cardiorespiratory fitness than moderate-intensity continuous training (MICT) in both healthy individuals and patients with cardiovascular diseases [10,11,12,13]. Exercise studies have shown positive effects on fatigue in cancer patients [14] and patients suffering from chronic fatigue syndrome (CFS) [15]. One reason for not including high-intensity exercise has been the risk of worsening fatigue, which may lead to high drop-out rates [7], but a study of patients with CFS showed that one session of HIIT did not aggravate fatigue more than one session of MICT [16]. To date, it remains unknown whether high-intensity exercise will affect fatigue differently compared to moderate-intensity exercise in patients with pulmonary sarcoidosis. Therefore, the main aim of this study was to investigate whether a single session of HIIT would affect sarcoidosis-related fatigue differently from a single session of MICT. The second aim was to evaluate the feasibility of an HIIT session in patients with sarcoidosis with the following outcomes: (1) completion of the entire session with four repetitions of 3 min; (2) adherence to the target heart rate (HR) and perceived exertion; (3) events during the session.

## 2. Material and Methods 

### 2.1. Study Design and Subjects

The study had a crossover design with a convenience sample of patients with pulmonary sarcoidosis recruited from LHL Hospital Gardermoen, a national pulmonary rehabilitation (PR) clinic in Norway. Patients (>18 years) with pulmonary sarcoidosis diagnosed in accordance with accepted guidelines [1], who attended a four-week exercise-based PR between April 2016 and June 2017 were eligible for this study. Patients were excluded if they (1) had a concurrent and predominant diagnosis of another significant respiratory disorder (asthma, chronic obstructive pulmonary disease (COPD), cystic fibrosis, or lung carcinoma); (2) had unstable cardiovascular disease; (3) were not able to perform the required physical tests and exercise training sessions because of co-morbidities. All patients were in a stable phase of the disease, and those on medication continued using their standard medication (steroids and methotrexate). The Regional Committee for Medical and Health Research Ethics approved the study (2014/2020), and written informed consent was obtained from each study participant. The study was registered at the ClinicalTrials.gov (NCT02735161) before the first patient was included. 

### 2.2. Background Variables

Information about medical history was collected from the pulmonary physician’s medical report. Body composition and lung function tests were performed according to international guidelines [17] and reference values [18]. Maximal exercise capacity (peak oxygen uptake, VO_2_peak) was assessed by a cardiopulmonary exercise test (CPET) (Ganshorn Schiller CS-200/ Vyntus CPX) on a treadmill, using a modified Bruce Protocol with reference values from a Norwegian population [19]. Submaximal exercise capacity was assessed by the 6 min walk test (6MWT) in accordance with standard criteria [20]. Fatigue was assessed using the Fatigue Assessment Scale (FAS). FAS is validated in patients with sarcoidosis [21,22] and consists of 10 items: five questions reflecting physical fatigue and five questions reflecting mental fatigue on a categorical response from 1 to 5. The total score range is from 10 to 50 points, where the cut-off for fatigue is >22 points [21]. All background data were collected at the first or second day of the PR program, and all patients responded to the questionnaires before the exercise tests.

### 2.3. Exercise Sessions 

All 41 patients in this crossover study performed two supervised exercise sessions on a treadmill (Technogym Jog 500, Technogym S.p.A, Cesena, Emilia Romagna, Italy), i.e., one HIIT session and one MICT session. As the patients primarily were participants in a four-week PR program, the choice of two sessions separated by a week was due to practical considerations. The HIIT session was performed the first week, and the MICT session was performed the second week. The two exercise sessions were conducted at the same time of the day to avoid the influence of individually daily variations of fatigue. The HIIT session consisted of a 6 min warm-up, four intervals of 3 min (4 × 3 min), active pauses of 2 min between each interval, and a 2 min cooldown with a total duration of 26 min. The target intensity of the 3 min intervals was >85% of peak heart rate (HRpeak) based on the obtained HRpeak from the CPET and/or perceived exertion of breathlessness ≥5 (severe) on the Borg CR10 scale [23]. During exercise, breathlessness was assessed, and HR was monitored using a sport watch (Polar V800, Polar Electro, Kempele, Finland). Speed and/or elevation were adjusted during the 3 min intervals to achieve 85% of HRpeak or Borg CR10 ≥ 5. The 2 min active pauses and the cooldown period consisted of either walking or jogging at an intensity corresponding to 3 (moderate) on the Borg CR10 scale.

The target intensity of the MICT session was 70% of HRpeak; HR was monitored and controlled during the entire session by a Polar sport watch. To keep the intensity constant, the treadmill function “Constant Pulse Rate” was used. The treadmill then automatically adjusted speed and/or elevation during the session to maintain the target intensity of 70% of HRpeak. Equal energy expenditure (kcal) was used to equate the HIIT and MICT sessions, and warm-up was included in both sessions, estimated by the Polar V800 watch. The MICT session lasted until the patient had consumed the same amount of individual kcal as in the HIIT session. The Polar V800 “Smart calories” function is based on the following individual parameters: gender, date of birth, bodyweight, height, HRpeak, resting HR, VO2peak, and a grading of how hard/often they usually exercise (hours per week). The Polar V800 has shown to be the most accurate sport watch for estimating kcal during aerobic activities in healthy individuals [24]. The wash-out time for fatigue as a response to a single exercise session is, to our knowledge, not known. To avoid carry-over effects of fatigue from other exercise sessions and physical activities in the PR program, the patients were not allowed to perform strenuous exercise 48 h before and 24 h after both sessions. The two sessions were supervised by a physiotherapist/project coordinator.

### 2.4. Outcome Variables

#### 2.4.1. Fatigue 

Several studies have failed to identify physiological biomarkers which correlate with sarcoidosis-related fatigue as a response to exercise in sarcoidosis [25,26,27], so the unidimensional Visual Analogue Fatigue scale (VAS-F) was found to be the most appropriate measure of fatigue in this study. The scale ranges from 0 to 100 mm, 0 indicates no fatigue, and 100 extreme fatigue. Fatigue was measured one minute before the exercise sessions (T0), one minute after the exercise sessions were completed (T1), and 24 hours after the sessions were completed (T2). The patients were each time asked to immediately report their perceived sarcoidosis-related fatigue. This scale has shown good reliability over 1–2 days [28] and sensitivity to changes in patients with interstitial lung disease (ILD) [29] and rheumatoid arthrosis (RA) [30]. The minimal clinically important difference (MCID) of change in VAS-F of 10 mm was established in patients with RA [31]. 

#### 2.4.2. Other Variables

To monitor the intensity of the two exercise sessions, HR and perceived breathlessness were used. HR was monitored continuously using the sport watch Polar V800. Breathlessness was assessed with the Borg CR10 scale [23]. This is a nonlinear category-ratio scale anchored between 0 (no exertion) to 10 (extreme), where 3 correspond to “moderate”, and 5 to “severe”. HR and breathlessness from the HIIT session correspond to the mean values in the second and third min during the 4 × 3 min intervals, and the mean values in every third min during the entire MICT session. Blood lactate was assessed by capillary puncture on a fingertip and was taken before, immediately after, and 24 h post-exercise, and immediately analyzed with a blood gas analyzer (ABL 800 Flex, Radiometer).

### 2.5. Statistical Analyses 

Power calculation was based on a change in MCID of 10 mm for VAS-F [31], an alpha value of 0.05, and a power value of 0.8. This led to a need for inclusion of 40 participants; *p* values of <0.05 were considered as statistically significant. All relevant variables were tested for normal distribution by visual inspection of the histograms, Q-Q plots, and test of normality. Because of the cross-over design, paired sample t-tests were used to detect statistically significant changes in fatigue from T0 to T1 and from T0 to T2 within and between the HIIT and the MICT sessions. All statistical analysis was performed with SPSS version 22 (SPSS Inc, Chicago, IL, USA).

## 3. Results

### 3.1. Flowchart and Baseline Characteristics

Figure 1 presents the flow chart of the study. Forty-seven of the 59 patients with pulmonary sarcoidosis who attended PR during the recruitment period met the inclusion criteria. Four declined to participate, and 43 patients were included. Two patients were excluded after one week because of relocation to other hospitals for further medical investigations, leaving 41 patients for the final analysis. 

The sample was evenly divided in females and males with normal lung function and slightly reduced exercise capacity. Thirty-nine of the 41 patients (95%) had fatigue FAS score > 22 points (Table 1).

### 3.2. Fatigue

No statistically significant differences in VAS-F scores were found between HIIT and MICT, neither at T1 (3.6 (13.5) mm vs 1.4 (13.5) mm, *p* = 0.326) nor at T2 (8.2 (17.0) vs 2.1 (17.1), *p* = 0.106). VAS-F increased slightly following both the HIIT and the MICT exercise session, with a statistically significant increase in VAS-F from 22.6 (18.8) mm to 30.9 (21.9) mm, *p* = 0.003 only at T2 after the HIIT session (Table 2). 

### 3.3. Feasibility of HIIT 

All 41 patients were able to complete the 26 min HIIT session. The target intensity of 85% of HRpeak was obtained by 33 of 41 patients (80%), and the perceived exertion of breathlessness Borg CR10 > 5 was obtained by 40 of 41 patients (98%) (Figure 2). One patient was not able to reach either 85% of HRpeak or Borg CR10 score of breathlessness of 5. A sub-group analysis showed that there was no statistically significant difference in fatigue, measured with the FAS, between the 33 patients who achieved the target intensity of >85% of HRpeak (FAS 30 (6) points) and the eight who did not achieve 85% of HRpeak (FAS 31(7) points), *p* = 0.550.

The intentional equal energy expenditure between HIIT (185 (65) kcal) and MICT (187 (65) kcal) was met, (*p* = 0.280) (Table 3). The mean Borg CR10 score and HR and lactate levels measured immediately after the sessions were significantly higher for the HIIT compared to the MICT (*p* < 0.0001) (Table 3). No adverse events occurred either during the HIIT or the MICT session. 

## 4. Discussion

To our knowledge, this is the first study that has examined the development of sarcoidosis-related fatigue after two exercise sessions with different intensities but with the same total amount of energy expended in patients with pulmonary sarcoidosis. There was no statistically significant difference in fatigue levels between one session of HIIT and one session of MICT, either immediately after (T1), or 24 h after exercise (T2). Our findings are in line with a comparable study by Sandler et al. [16] who compared one session of HIIT and one of MICT in 14 patients with CFS. They also found no statistically significant difference in fatigue between the two sessions. One of the previous arguments against HIIT in patients with sarcoidosis has been the development of fatigue [7]. Both these studies contradict these findings, however, by showing that a single session of HIIT does not aggravate fatigue more than a single session of MICT in patients suffering from fatigue.

When exploring the effects on sarcoidosis-related fatigue in the context of exercise, we have to be aware of the influence of acute exercise-induced fatigue due to physiological stress which increases with increasing exercise intensity [32]. The patients were carefully informed that the focus of self-reported fatigue was sarcoidosis-related fatigue, and not exercise-induced fatigue or peripheral muscle fatigue. In our clinic, we have experienced that patients with fatigue are able to distinguish between sarcoidosis-induced fatigue and exercise-induced fatigue. This is, as expected, confirmed by the statistically significant higher rating of perceived exhaustion of breathlessness (Borg CR10), HR, and blood lactate levels during the HIIT session compared to the MICT session. However, the perception of fatigue did not show any statistically significant difference between the two sessions either immediately after or 24 h after. This is also confirmed by the lack of association between perceived fatigue and exercise intensity, in relation to breathlessness, HR, and blood lactate (data not presented) and suggests that breathlessness, HR, or blood lactate per se may not be the best measures to use as indicators of post-exercise fatigue. The post-exercise measure points in this study were based on a former study and feedback from our patients reporting the onset of acute fatigue on the following day [25]. In this study, a trend toward an increase in fatigue was seen after both the HIIT and the MICT session, with a statistically significant increase 24 h after the HIIT session only. Our observations are in keeping with the study of Sandler et al. [16], showing an increase in fatigue following both the HIIT and the MICT session, even up to 96 hours post-exercise in CFS patients. Both studies show the clinical importance of having several measure points to capture the development of fatigue as a response to exercise. Patients report the onset of fatigue after several hours to be more frustrating than the acute onset, as might be expected after an exercise session. Therefore, the clinical implications of our findings might give the patients better self-efficacy in managing post-exercise fatigue. However, it is important to note that the changes in fatigue shown by this study, including the statistically significant increase of 8.2 mm after HIIT, are not considered to be clinically significant as they are below the MCID of 10 mm [31]. In light of the non-statistically or clinically significant changes of fatigue obtained in our study, it is relevant to discuss the sensitivity of the VAS-F. The VAS-F has been used in another study to measure fatigue as an acute response to exercise in sarcoidosis [25] and has shown good reliability and sensitivity to changes in fatigue [28,29]. Thus, we considered VAS-F to be the most appropriate scale to use per se.

Because of the need for establishing an optimal training program (mode of exercise, duration, and intensity) for patients with pulmonary sarcoidosis [33], the second aim of this study was to evaluate the feasibility of HIIT as an alternative to traditionally moderate-intensity programs [5,6,7]. The target intensity of HIIT in the present study was defined as >85% of HRpeak, in keeping with recommendations for high-intensity training [34]. Previous exercise studies of patients with sarcoidosis identified the planned target intensity of heart rate or perceived exertion on Borg scales but did not report if the patients actually achieved it during exercise interventions [5,6,7,8]. To our knowledge, this is the first study that demonstrates that patients with sarcoidosis actually manage to achieve and maintain the target intensity during an HIIT session. The adherence to the intended intensity in this study was good. In fact, 80% of the patients achieved ≥85% of HRpeak, and 40 out of 41 reported a Borg CR10 score of ≥5. In addition, all 41 patients managed to complete the entire HIIT session without any events. This indicates that HIIT might be feasible in an exercise program for patients with sarcoidosis. Three of the existing exercise studies in patients with sarcoidosis followed protocols of moderate-intensity training (50–60% of peak work) [5,6,7]. Although the patients seemed to improve in both exercise capacity and fatigue, the changes were small. In a pilot study by Strookappe et al. [6], only 6/12 patients with sarcoidosis had an improvement of 10% in 6MWD, and only 4/12 showed an improvement in fatigue. In the study by Marcellis et al. [7], 9/18 patient achieved 5–10% improvement in 6MWD, while fatigue was improved by 4 points in 6/18 and 10% in 9/18 patients. As exercise intensity is one of the key factors to improve exercise capacity [34], the moderate-intensity protocols in the above-mentioned studies may be the reason for the small improvements in exercise capacity and potentially small improvements in fatigue. Deconditioning has been proposed to be a contributing factor leading to fatigue [35], and exercise capacity has shown to be significantly associated with fatigue [36]. A goal when treating patients with sarcoidosis could therefore be to improve exercise capacity. As reduced exercise capacity is present in patients with sarcoidosis [4], it is possible that the implementation of HIIT, which is considered most effective in order to improve exercise capacity, could lead to a more significant reduction of fatigue. In addition, HIIT has shown to be more time-efficient and enjoyable compared to MICT [37,38]. This might have certain important clinical implications in terms of exercise adherence. 

A review recommends the exercise intensity for patients with sarcoidosis to be personalized and adjusted for daily fluctuations of fatigue [33]. The sample in this study had a mean fatigue score of 30 (6) points (FAS), which is similar to scores determined by other exercise studies of patients with sarcoidosis of FAS ± 30 points [5,6,7]. Our sub-analysis showed no statistically significant difference in FAS scores between patients who achieved 85% of HRpeak and those who did not during the HIIT. Our findings indicate that the initial level of fatigue does not affect the patient’s ability to perform high-intensity exercise training. We agree that exercise intensities in sarcoidosis should be personalized, but on the basis of individual preferences and other considerations, rather than on the basis of the level of fatigue in particular. The burden of comorbidities in sarcoidosis might be relevant when prescribing exercise programs. Cardiovascular comorbidity is reported to be the most prevalent comorbidity in patients with sarcoidosis [39]. In addition, patients with one or more comorbidities show a forceful reduction in physical activity compared to patients without comorbidities [40]. This highlights the importance of initiating physical exercise in this population, both to prevent comorbidities and to improve fatigue.

The HIIT protocol of 4 × 3 min in this study is a modified version of a 4 × 4 min protocol which has shown to be feasible for patients with several other diseases, such as coronary heart diseases, metabolic syndrome, and heart failure [41,42,43]. Over the past several years, we have gained a great deal of experience with the use of both the 4 × 4 min and the modified 4 × 3 min protocols in patients with different lung diseases. The patients have shown good compliance, and the transferability of the HIIT protocol to other modes of exercises is good. Patients have used the protocol when walking outdoor, both uphill and with increased walking speed on level ground, and on treadmill, stationary cycle, rowing machine, and elliptical cross trainer. This flexibility makes these protocols easy to transfer to available equipment at home-based settings as well, which may increase the long-term adherence to regular exercise for patients with pulmonary sarcoidosis.

One method used when comparing two exercise sessions with different intensity is to equate the total work performed by energy expenditure (kcal) [41,42,43]. To pre-define the time of each session, the calculation is depended of the average VO2peak [41]. The sample in the current study was heterogeneous in regard to VO2peak (highest: 3.26 L/min and lowest: 0.94 L/min), leading to the same wide range of energy expenditure after the exercise sessions (280 kcal and 69 kcal, respectively). Taking into account the crossover design, the individual kcal consumption measured by a sport watch, not the pre-defined time, was considered as the most appropriate method to equate the two sessions.

### Strength and Limitations

Because of the small numbers of patients with sarcoidosis attending PR in Norway with an annual number of 30, a cross-over study was considered as the most appropriate and feasible design. Both sessions were supervised, and regular monitoring of heart rate and perceived exertion ensured the target intensity was reached and contributed to quality assurance of the results. The location of an in-patient PR clinic made it possible to customize the project schedule in relation to the PR schedule, to avoid inflicting additional stress on the participants. However, there are some limitations to this study. Firstly, the absence of a random allocation. Randomization was considered but found difficult to implement, as the two sessions were matched by energy expenditure. If randomized, the MICT protocol must have had a defined duration as well. The caloric consumption after a fixed duration of the MICT session could have resulted in an interruption of the HIIT before completing the 26 min, due to the challenges described in the section above. As feasibility of HIIT was a secondary aim, we considered a random allocation not to be suitable. Secondly, as daily variation of fatigue is reported in patients with sarcoidosis, this could potentially have influenced our results. Measures of VAS fatigue on days without exercise could have been added to control for this bias. Thirdly, the design with only one session of HIIT is a limitation to predict the long-term effects of high-intensity exercise training on fatigue. Finally, since we do not know how exercise of any modalities affects the immunological, muscular, or respiratory functions in sarcoidosis, we cannot preclude that there are differential effects of MICT and HIIT on these systems.

## 5. Conclusions

The results from this study show that patients with pulmonary sarcoidosis are able to safely perform a single session of HIIT without worsening of sarcoidosis-related fatigue. The change in fatigue was comparable to those seen following an MICT session. These preliminary findings support the need for further research on the long-term effects of HIIT on fatigue in patients with sarcoidosis. 

## Figures and Tables

**Figure 1 jcm-08-00460-f001:**
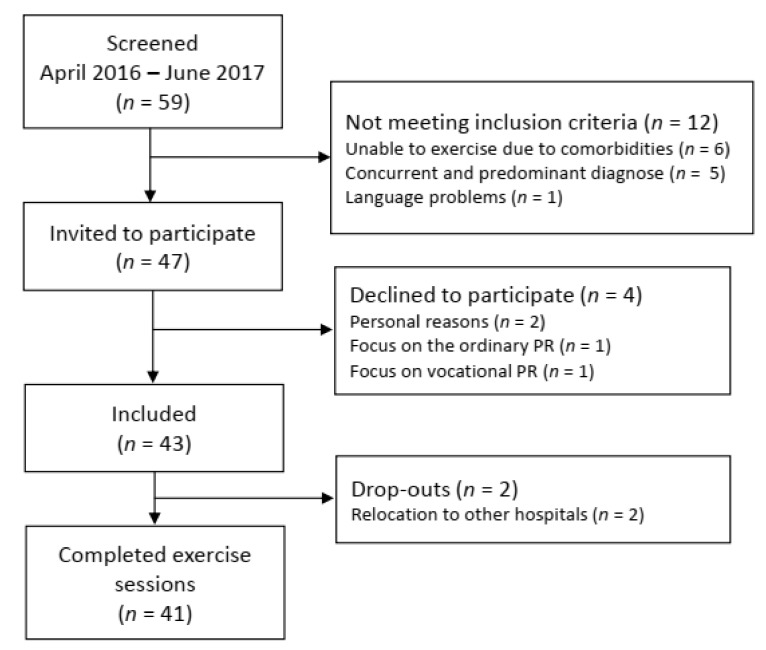
Flowchart of recruitment, inclusion, and drop-outs. PR: pulmonary rehabilitation.

**Figure 2 jcm-08-00460-f002:**
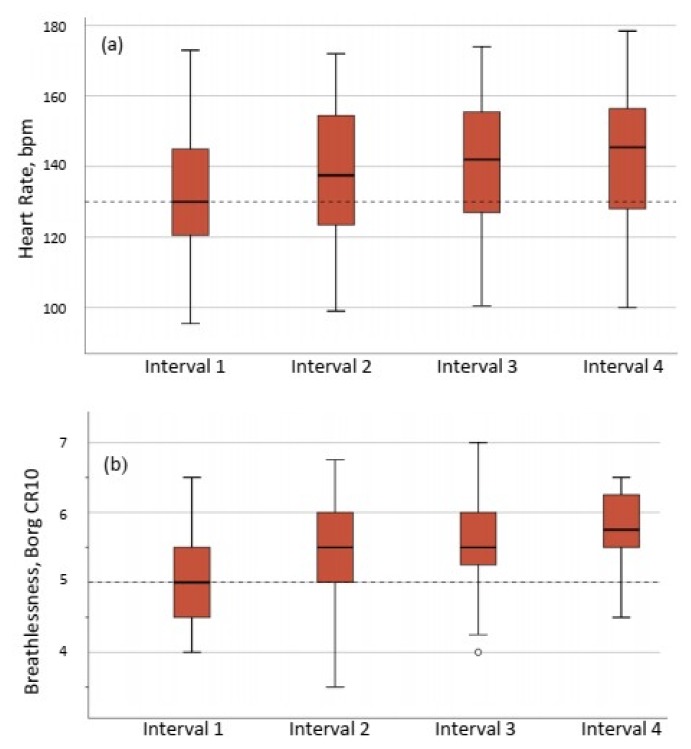
Exercise intensity during the four intervals of the HIIT session expressed as (**a**) percentage of peak heart rate, and (**b**) perceived breathlessness assessed by Borg CR10 scale (range 0–10). The horizontal dotted lines show target intensity, (2b) ◦ = outlier, (**a**) 85% of HRpeak and (**b**) Borg CR10 = 5.

**Table 1 jcm-08-00460-t001:** Baseline characteristics.

Characteristics	*n* = 41
Gender (M/F)	20/21
Age (years)	53 (11)
BMI (kg/m^2^)	30 (6)
FVC (% predicted)	93 (21)
FEV_1_ (% predicted)	82 (22)
TLC (% predicted)	93 (17)
DLCO (% predicted)	96 (17)
VO_2peak_ (mL· kg^−1^·min^−1^)	24.6 (6.8)
VO_2peak_ (% predicted)	72 (19)
6MWD (meter)	580 (81)
Fatigue, FAS (points)	30 (6)
Medication	
Prednisolon (n (%))	11 (27)
Methotrexate (n (%))	6 (15)

Data are presented as mean (SD) or *n* (%). BMI: body mass index, FVC: forced vital capacity, FEV_1_: forced expiratory volume in 1 s, TLC: total lung capacity, DLCO: diffusion capacity of the lung for carbon monoxide, VO2peak: peak oxygen uptake, 6MWD: 6-min walking distance, FAS: fatigue assessment scale (10–50 points).

**Table 2 jcm-08-00460-t002:** Change in fatigue (VAS-F) within and between HIIT and MICT sessions.

		VAS-F		VAS-F from T0 to T1			VAS-F from T0 to T2		
	T0	T1	T2	Mean Change	∆ Group Diff.		Mean Change	∆ Group Diff.	
	Mean (SD)	Mean (SD)	Mean (SD)	Mean (SD)	Mean (SD)	*p*-Value	Mean (SD)	Mean (SD)	*p*-Value
HIIT	22.6 (18.8)	26.2 (20.7)	30.9 (21.9)	3.6 (13.5)			8.2 (17.0) *		
MICT	26.9 (23.7)	28.3 (21.4)	29.0 (21.6)	1.4 (13.5)	2.2 (14.3)	0.326	2.1 (17.1)	6.1 (23.8)	0.106

All data presented as mean (SD). VAS-F: visual analogue fatigue scale, 0–100 mm, T0: before the training session, T1: immediately after the training session, T2: 24 h after the training session, Group Diff.: group difference, HIIT: high-intensity interval training, MICT: moderate-intensity continuous training, * *p* = 0.003.

**Table 3 jcm-08-00460-t003:** Exercise responses and duration for HIIT and MICT sessions.

	HIIT *n* = 41	MICT *n* = 41	*p*-Value
Energy expenditure, kcal	185 (65)	187 (65)	0.28
Breathlessness, Borg CR10	5.8 (0.6)	3.1 (0.8)	<0.0001
Heart rate, %HR_peak_	90 (8)	73 (6)	<0.0001
Blood lactate, mmol·L	5.8 (2.7)	2.2 (0.8)	<0.0001
Time, min:s	26.00	37:43	<0.0001

Data are presented as mean (SD). HIIT: high intensity interval training, MICT: moderate intensity continuous training.

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
