# Peer review of "The Effects of High- Versus Moderate-Intensity Exercise on Fatigue in Sarcoidosis"

_jcm, 2019, doi:10.3390/jcm8040460_

Round 1
Reviewer 1 Report
Nicely written! A few minor comments:
1) Figure-1: Please improve the figure quality. It looks little blurry and small font.
2) Table-1: Table looks like an image (cropped). Could you please replace it?
3) Figure-2: Needs to be improved, in terms of font.
Thank you!
Author Response
Thank you for a thorough and valuable review of our paper. Each point has been thoroughly studied and has been of great help in our work with improving the figures and table in the manuscript.

Reviewer 2 Report
I have these general comments: 1. Firstly the investigation has been conducted with a good sample size for this population. The study appears to be novel for this population, so has some contribution to the existing literature 2. Assuming the exercise trial ie moderate vs high intensity were energy expenditure matched you would assume thelevel of local peripheral muscle fatigue was similar which is what has been shown. It is merely that perhaps fatigue as determined by VAS manifests in another systems ie central or respiratory muscle which is not determined. 3. It is uncertain whether the fatigueresponse observed is unique to the population or whether the measurements obtained were just not sensitive to determining whether there is residual fatigue 4. (Considering point 3) The limitations are no age matched ‘healthy control’. The subjective nature of VAS without any other physiological or in fact immunological measurements. I would have thought pulmonary muscle fatigue might be relevant and easily measured. I can also see value in determining some blood markers reflective of acute stress response/immune function eg CK, cortisol, TNF-a, etc
Author Response
Thank you for a thorough and valuable review of our paper. Each comment has been thoroughly studied and has been of great help in our work with improving the manuscript.

Reviewer 3 Report
The aim of this study is to compare the effect of high vs. moderate intensity exercise on fatigue in sarcoidosis. Forty one patients completed two training sessions of HIIT and MICT. The Visual Analoug Scale was used to compare changes in fatigue. Authors concluded that a single session of HIIT did not affect fatigue differently than a single session of MICT.
I have significant specific comments listed below:
Authors should clearly define what the purpose of the study was. In my opinion you analysed the differences of fatigue in response to high v. moderate exercise (Title of the manuscript) after rehabilitation program but not the effect of trainning (HIIT vs. MICT) (page 1 Line 21-22.
A limitation of the present study is the lack of a pre-training resuts of VAS-F after the HII and MIC exercise session (but not training).
Characteristics of the methods used in Materials and Methods Section should be precisely described. The fatigue score presented in the table 1 as a FAS points should be defined. Description of the measurments is not clear. What was the reasen to present tha FAS score before the PR but not after the PR.
Although the training protocol adopted a crossover design that was separated by one week detreining period, whether one-week period can wash out training effcts remain unclear. This study should include both pretraining and detraining (pre PR and post PR) measurments of fatigue but in my opinion not only VAS-F scale but also FAS and Borg scale (Borg CR10score page 6 line 193).
What about the somatic (obesity) and physical performance differences of the subjects, this is not clear and could affect the findings dramatically.
It is not clear why the individual kcal achievent was used for the time of MICT sessions. Please claryfy (some references must be added).
It would be very good if you could analyze the effect of physical activity of sarcoidosis patients on prevention fatigue according to literature data (Introduction Section, line 39-45). The Authors need to explain these results suggested by previous authors in more details especity abut the patients exercise tolerance and the effect of comorbidities in Discussion section. The sentence „only four studies…” is wrong. Some references could be added.
Jastrzębski D., et al. Fatigue in sarcoidosis and exercise tolerance, dyspnea, and quality of life. Advances in Experimental Medicine and Biology. 2015, 833; 31-36.
Pilzak K., A. et al. Physical functioning and symptoms of chronic fatigue in sarcoidosis patients. Advances in Experimental Medicine and Biology-Neuroscience and Respiration
Kostorz S., et al. Predominance of comorbidities in the detriment of daily activity in sarcoidosis patients. Advances in Experimental Medicine and Biology-Neuroscience and Respiration. 2018; 37: 7-12.
Please, explain, the association between the perception of fatigue, HR and blood lactate levels in response to both exercise protocols.
Please, comment the clinical significance of measurments of fatigue and physiological variables, or modify your interpretation.
Author Response

(The authors gave the same response as above.)

Round 2
Reviewer 2 Report
The authors have addressed the minor considerations. The manuscript has been adjusted according to the other reviewers comments